# Structural and functional insights into IZUMO1 recognition by JUNO in mammalian fertilization

Kazuki Kato[1,*], Yuhkoh Satouh[2,*], Hiroshi Nishimasu[1,3,*], Arisa Kurabayashi[1], Junko Morita[1], Yoshitaka Fujihara[2], Asami Oji[2], Ryuichiro Ishitani[1], Masahito Ikawa[2] & Osamu Nureki[1]

Sperm–egg fusion is the critical step in mammalian fertilization, and requires the interaction between IZUMO1 on the sperm surface and JUNO (also known as folate receptor (FR) 4 or IZUMO1R) on the egg surface. Whereas other FRs bind and uptake folates, JUNO binds IZUMO1 and establishes the cell–cell adhesion. However, the mechanism of IZUMO1 recognition by JUNO has remained elusive. Here we report the crystal structure of mouse JUNO, at 2.3 Å resolution. A structural comparison of JUNO with the FRs revealed that JUNO and the FRs have similar overall structures, but JUNO lacks the folate-binding pocket, thereby explaining the inability of JUNO to bind folate. Further complementation of *Juno* knockout eggs with mutant *Juno* messenger RNAs revealed that the conserved, surface-exposed tryptophan residue of JUNO is required for sperm binding and fertilization. Our structure-based *in vivo* functional analyses provide a framework towards a mechanistic understanding of mammalian gamete recognition.

[1] Department of Biological Sciences, Graduate School of Science, The University of Tokyo, 2-11-16 Yayoi, Bunkyo-ku, Tokyo 113-0032, Japan. [2] Research Institute for Microbial Diseases, Osaka University, 3-1 Yamadaoka, Suita, Osaka 565-0871, Japan. [3] PRESTO, JST, 2-11-16 Yayoi, Bunkyo-ku, Tokyo 113-0032, Japan. * These authors contributed equally to this work. Correspondence and requests for materials should be addressed to M.I. (email: ikawa@biken.osaka-u.ac.jp) or to O.N. (email: nureki@bs.s.u-tokyo.ac.jp).

Among many factors involved in sperm–egg fusion[1], IZUMO1 (ref. 2) on the spermatozoon and the IZUMO1 counter-receptor JUNO[3] on the egg are the only factors proven to form an intercellular bridge, and the deletion of either gene leads to the failure of gamete membrane fusion. Consistent with the ability of a hamster egg to fuse with human sperm[4], hamster JUNO can bind human IZUMO1 (ref. 5), highlighting the importance of the IZUMO1–JUNO interaction in cross-species gamete recognition. IZUMO1 is a type-I transmembrane protein, comprising an ectodomain and a single membrane-spanning region[2]. The IZUMO1 ectodomain is composed of an immunoglobulin-like domain and an Izumo domain, which can be further divided into an N-terminal unstructured region and an α-helical core region important for sperm–egg binding[6]. JUNO is a glycophosphatidylinositol (GPI)-anchored cysteine-rich glycoprotein, and was originally identified as folate receptor (FR) 4, which shares 52% sequence identity with FR1 and FR2 in mice[3]. However, JUNO binds IZUMO1 and is involved in the sperm–egg adhesion[3], whereas the FRs bind folate and participate in folate uptake[7–10]. The crystal structures of FR1 and FR2 revealed that the FRs adopt a globular architecture stabilized by multiple disulfide bonds and recognize folate in a folate-binding pocket[11,12]. Consistent with the distinct specificities between JUNO and the FRs, the amino-acid residues forming the folate-binding pocket are strictly conserved in the FRs, but not in JUNO. Recently, the crystal structure of mouse JUNO was reported[13]. The structure revealed conformational differences in three loop regions between JUNO and the FRs, despite they have similar overall structures. In addition, an avidity-based extracellular interaction screen suggested that the flexible loop regions of JUNO are involved in IZUMO1 binding[13]. However, the molecular mechanism by which JUNO specifically recognizes IZUMO1 and participates in the sperm–egg adhesion step remains to be fully understood. To gain mechanistic insights into the IZUMO1 recognition by JUNO, we solved the crystal structure of the N73D mutant of mouse JUNO and performed the structure-based mutational analyses.

## Results

**Crystal structure of the N73D mutant JUNO ectodomain**. We prepared the wild-type (WT) ectodomain of mouse JUNO (residues 20–221), using the baculovirus-Sf9 expression system (Fig. 1a and Supplementary Fig. 1a). We performed crystallization screens, but failed to obtain crystals of the WT JUNO ectodomain. Mouse JUNO has two putative N-glycosylated sites (Asn73 and Asn185; Fig. 1a), and the glycosylation may hinder the crystallization. This notion is consistent with the fact that the WT ectodomain of mouse JUNO (residues 20–221), which was expressed in human embryonic kidney 293S (HEK293S) GnT1⁻ cells and enzymatically deglycosylated, was successfully crystallized[13]. The N185D mutant JUNO ectodomain was not expressed as a soluble protein in insect cells, suggesting that the Asn185-linked glycan contributes to the protein stability. In contrast, the N73D mutant JUNO ectodomain was expressed at a similar level to that of the WT JUNO ectodomain. We thus crystallized the N73D mutant ectodomain of mouse JUNO (residues 20–221), and determined its crystal structure by molecular replacement at 2.3 Å resolution (Fig. 1b, Table 1 and Supplementary Fig. 1b). The structure revealed that JUNO adopts a globular architecture comprising two long α helices (α3 and α4) surrounded by five short α helices (α1, α2 and α5–α7), a two-stranded anti-parallel β-sheet (β2 and β3), a two-stranded parallel β-sheet (β4 and β7) and a mixed three-stranded β-sheet (β1, β5 and β6) (Fig. 1b). The overall structure is stabilized by eight conserved disulfide bonds (Fig. 1b). The Asn185-linked glycan interacts with the side chains

of Asn146 and Leu148 (Fig. 1c), consistent with the reduced stability of the N185D mutant.

Recently, the crystal structure of the WT ectodomain of mouse JUNO (PDB code 5EJN) was determined at 2.7 Å resolution[13]. The crystal of WT JUNO belongs to the space group $P2_1$, with two protomers in the asymmetric unit (Mol A and Mol B) (Fig. 1d), whereas our crystal of N73D mutant JUNO belongs to the space group $P2_12_12$, with one protomer in the asymmetric unit. Our structure of N73D mutant JUNO is essentially identical to that of WT JUNO (PDB code 5EJN)[13], with root mean square deviations of 0.74 and 0.70 Å for Mol A (157 Cα atoms) and Mol B (166 Cα atoms), respectively (Fig. 1e), suggesting that the N73D mutation did not substantially affect the overall structure. A comparison between the three structures revealed that JUNO comprises a rigid core and three surface-exposed, flexible regions (flexible regions 1–3) (Fig. 1e). In our N73D mutant JUNO structure, flexible regions 1 and 3 adopt loop conformations (flexible region 3 is partly disordered), while flexible region 2 forms a β-hairpin (Fig. 1b). In contrast, flexible regions 1–3 are largely disordered in Mol A, whereas flexible regions 1 and 3 are disordered and flexible region 2 forms a β-hairpin in Mol B of the WT JUNO structure (Fig. 1d)[13]. In the WT JUNO, Asn73 on flexible region 2 is disordered and does not interact with the protein (Fig. 1d), consistent with our data showing that the N73D mutation does not affect the protein expression.

**Structural comparison of JUNO with the FRs**. Mouse JUNO shares sequence and structural similarities with human FR1 (ref. 11) (PDB code 3LRH, 54% sequence identity, root mean square deviation of 1.0 Å for 176 Cα atoms) and human FR2 (ref. 12) (PDB code 4KMZ, 54% sequence identity, root mean square deviation of 1.2 Å for 179 Cα atoms) (Fig. 2a–c). The eight disulfide bonds are also conserved in JUNO and the FRs (Fig. 2a–c). In FR1 (ref. 11) and FR2 (ref. 12), the folate is bound to the folate-binding pocket formed by the core helices and loop regions, and is extensively recognized by the protein (Fig. 2d,e). In FR1, Tyr85/Trp171 and Tyr60/Trp102 form stacking interactions with the pterin and aminobenzoate rings of folate, respectively[11]. In addition, Asp81, Arg103, Arg106, His135 and Ser174 hydrogen bond with the pterin ring of folate[11] (Fig. 2e). The Tyr60, Asp81, Tyr85, Trp102, Arg103, Arg106, His135 and Trp171 residues of FR1, respectively correspond to the Phe72, Ala93, His97, Gly114, Gln115, Arg119, Leu148 and Trp184 residues of JUNO, which form a central pocket equivalent to the folate-binding pocket of FR1 (Fig. 2f,g). A structural comparison of JUNO with FR1 revealed that folate cannot be accommodated in the central pocket of JUNO (Fig. 2h). In particular, JUNO Trp184 adopts a different configuration from that of the equivalent FR1 Trp171, and would sterically clash with the folate (Fig. 2h). Furthermore, Asp81 and Arg103 in FR1 are replaced with Ala93 and Glu117 in JUNO, respectively (Fig. 2h). These structural differences explain the inability of JUNO to bind folate.

**Putative IZUMO1-binding surface**. To identify the IZUMO1-binding residues of JUNO, we generated *Juno* knockout (KO) mice using the CRISPR-Cas9 system (Supplementary Fig. 2), and examined whether the sperm-fusing ability of unfertilized eggs derived from *Juno* KO mice can be complemented by the injection of mRNA encoding the WT or mutants of mouse JUNO (Fig. 3a). As expected, the *Juno* KO eggs failed to fuse with the sperm, whereas the injection of mRNA encoding WT JUNO restored their sperm-fusing ability (Fig. 3b, Supplementary Fig. 3 and Table 2). N73D mutant JUNO rescued the sperm-fusing ability of the *Juno* KO eggs (Fig. 3b and Table 2), indicating that

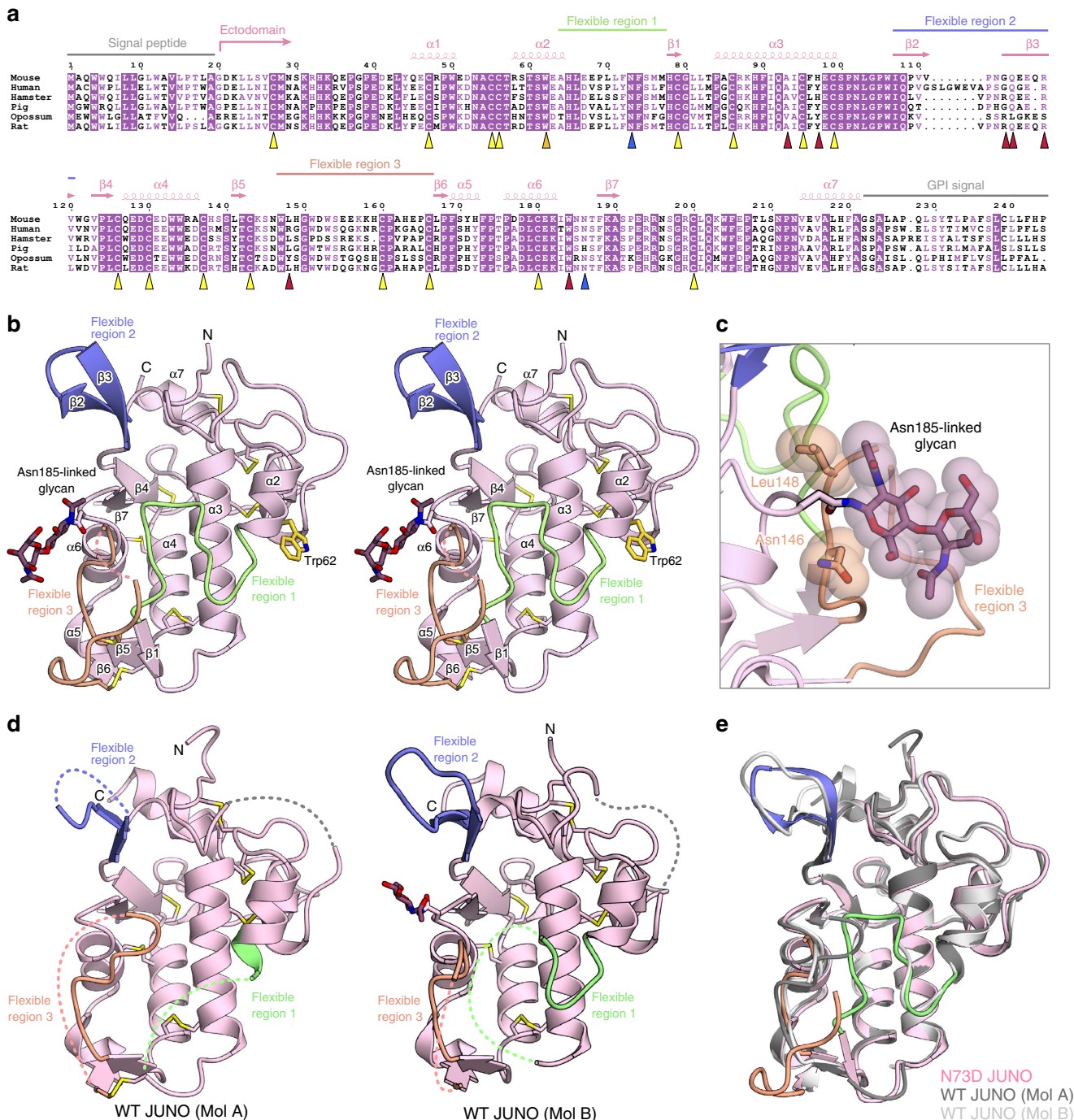

**Figure 1 | Crystal structure of JUNO.** (**a**) Multiple sequence alignment of JUNOs from different species. In mouse JUNO, the two glycosylation sites (Asn73 and Asn185) and the eight conserved disulfide bonds are indicated by blue and yellow triangles, respectively. Trp62 is indicated by a gold triangle. Residues forming the central pocket are indicated by red triangles. (**b**) Ribbon representation of the N73D mutant ectodomain of mouse JUNO (stereo view). Flexible regions 1–3 are coloured green, blue and orange, respectively. The disulfide bonds and N-linked glycans are shown as stick models, and the disordered regions are shown as dashed lines. Trp62 is shown as a gold stick. (**c**) Asn185-linked glycan. The Asn185-linked glycan and the side chains of Asn146 and Leu148 are depicted as stick and space-filling models. (**d**) Ribbon representations of the ectodomain of WT JUNO (PDB code 5EJN)[13]. Mol A (left) and Mol B (right) are shown as in **b**. (**e**) Superimposition of our N73D mutant JUNO (coloured) and WT JUNO (PDB code 5EJN) (grey)[13].

the Asn73-glycan is dispensable for the sperm–egg interaction. His97 and Trp184 in the central pocket are conserved among the JUNOs from different mammalian species (Figs 1a and 2g), implying the functional importance of the central pocket. However, both the H97A and W184A mutants restored the sperm-fusing abilities of the *Juno* KO eggs (Fig. 3b and Table 2), indicating that the central pocket is not involved in IZUMO1

binding. A close inspection of the conservation of surface residues revealed that the loop regions around the central pocket are less conserved, whereas the molecular surface opposite from the central pocket is relatively conserved among the JUNOs from different mammalian species (Fig. 3c). In particular, the hydrophobic Trp62 and Leu66 residues are exposed to the solvent (Fig. 3c). Notably, these residues are highly conserved among the

## Table 1 | Data collection and refinement statistics.

| | N73D mutant JUNO |
|---|---|
| *Data collection* | |
| Space group | $P2_12_12$ |
| Cell dimensions | |
| *a, b, c* (Å) | 54.4, 87.1, 42.4 |
| *α, β, γ* (°) | 90.0, 90.0, 90.0 |
| Resolution (Å) | 46.2-2.30 (2.44-2.30)* |
| $R_{sym}$ | 0.125 (0.818) |
| $I/\sigma I$ | 11.2 (2.02) |
| Completeness (%) | 99.7 (98.1) |
| Redundancy | 5.90 (5.58) |
| | |
| *Refinement* | |
| Resolution (Å) | 46.2-2.30 |
| No. reflections | 9,441 |
| $R_{work}/R_{free}$ | 0.207/0.235 |
| No. of atoms | |
| Protein | 1,541 |
| Ligand/ion | 14 |
| Water | 44 |
| *B*-factors (Å$^2$) | |
| Protein | 41.0 |
| Ligand/ion | 55.6 |
| Water | 37.2 |
| R.m.s. deviations | |
| Bond lengths (Å) | 0.003 |
| Bond angles (°) | 0.61 |
| Ramachandran statistics (%) | |
| Favoured | 98 |
| Allowed | 2 |
| Outlier | 0 |

*Highest resolution shell is shown in parentheses.

JUNOs (Fig. 1a), but not in the FRs (Fig. 2a), suggesting their functional importance. Indeed, the W62A mutant, but not the L66A mutant, failed to rescue the sperm-fusing ability of *Juno* KO eggs (0/21 and 15/18 eggs fertilized in W62A and L66A, respectively) (Fig. 3b and Table 2), suggesting that Trp62 is involved in the IZUMO1–JUNO interaction. Given that the α-helical core region (residues 57–113) of IZUMO1 is important for the interaction with JUNO[6], these results suggested that Trp62 of JUNO interacts with hydrophobic residues on the α-helical core region of IZUMO1.

**Involvement of JUNO Trp62 in the cell–cell adhesion.** Recent studies indicated that the interaction between JUNO and IZUMO1 is sufficient to generate the cell–cell interaction, but not cell–cell fusion[6,14]. We thus examined the interaction between HEK293T cells expressing mouse JUNO and mouse sperm expressing IZUMO1-mCherry (Red-IZUMO1)[15]. The acrosome-reacted spermatozoa that are distinguished by the distribution of Red-IZUMO1 signal on equatorial or postacrosomal region of sperm head[15] preferentially bound to the HEK293T cells expressing mouse JUNO, and the Red-IZUMO1 signals did not transfer and disperse onto the HEK293T cell surface (Fig. 4a). These results indicated that the sperm cells bind to the JUNO-expressing cells, but do not fuse with them. We next examined the interaction between HEK293T cells expressing the WT or mutants of mouse JUNO and transgenic spermatozoa with mitochondrial DsRed2, which exhibits strong fluorescence and thus enables us to count bound spermatozoa on the seeded cells (Fig. 4b and Supplementary Figs 4 and 5). Consistent with the results of our sperm–egg fusion assays (Fig. 3b), the HEK293T cells expressing the W62A JUNO mutant had reduced sperm-

binding ability, as compared to those expressing the WT or other mutants (Fig. 4b,c and Supplementary Fig. 5b). These results indicated that Trp62 of JUNO participates in the interaction with IZUMO1, rather than the subsequent fusion process.

**Interaction between human JUNO and mouse IZUMO1.** A previous study showed that human eggs can fuse with mouse sperm[14], suggesting that human and mouse IZUMO1 and JUNO are interchangeable. When we expressed human JUNO on *Juno* KO mouse eggs, we could not detect the expression by immunostaining, possibly due to the lack of antibody cross-reactivity (Supplementary Fig. 3). However, whereas *Juno* KO eggs failed to fuse the mouse sperm (0%; 0/21 eggs fertilized), *Juno* KO mouse eggs expressing human JUNO successfully fused with the mouse sperm (50.0%; 12/24 eggs fertilized) (Fig. 3b and Table 2). In addition, the mouse sperm bound to the HEK293T cells expressing human JUNO (Fig. 4b,c). These results indicated that human JUNO can interact with mouse IZUMO1, consistent with the previous report[14].

### Discussion

The crystal structure of N73D mutant JUNO described here, along with the recently reported WT JUNO structures[13], revealed that JUNO has a core fold and three surface-exposed, flexible regions. A previous functional analysis suggested that flexible regions 1 and 3 around the central pocket are involved in IZUMO1 binding[13]. Based on these results, Han *et al.* proposed that the α-helical region of IZUMO1 interacts with the surface groove surrounded by the two flexible loop regions of JUNO[13]. Our *in vivo* analyses revealed that Trp62, rather than the central pocket, is critical for sperm binding and fertilization, thereby highlighting the importance of Trp62 for IZUMO1 binding. Given the proximity of Trp62 to flexible regions 1 and 3, we suggest that JUNO interacts with IZUMO1 through the surface encompassing Trp62 and flexible regions 1 and 3 (Fig. 5). Importantly, Trp62 is strictly conserved among the JUNOs, whereas the flexible regions are relatively divergent (Fig. 1a). These observations suggest that Trp62 plays conserved roles in IZUMO1 binding, whereas the flexible regions contribute to IZUMO1 binding in species-specific manners. This notion is consistent with our data indicating that human JUNO can interact with mouse IZUMO, with reduced efficiency (Figs 3b and 4c and Table 2).

Previous studies showed that the cultured cells expressing mouse IZUMO1 can bind to eggs, but do not fuse with them[6,14,16]. Here we showed that the cultured cells expressing mouse JUNO can bind to spermatozoa, but do not fuse with them (Fig. 4c). In addition, a recent study revealed that JUNO is enriched in the sperm–egg interface before the beginning of membrane fusion but disappears a few minutes after fusion has started while part of IZUMO1 still present[16]. These results suggest that IZUMO1 has another receptor that could be involved in membrane fusion. The complementation assays presented in this study utilizing *Juno* KO oocyte and cultured cells would be usable even in the identification of the novel IZUMO1 receptor and/or in detailed studies of gamete recognition such as a species specificity. In summary, our structural and functional approach provide a framework towards a mechanistic understanding of sperm–egg fusion in mammalian fertilization.

### Methods

**Protein preparation.** The gene encoding the ectodomain of mouse JUNO (residues 20–221) was inserted into the modified pFastBac1 vector (Invitrogen), which contains the N-terminal signal peptide of Hemolin from *Hyalophora cecropia* (residues 1–18) and a C-terminal HRV3C protease cleavage site followed by a 10 × His tag. The putative glycosylation sites (Asn73 and Asn185) were mutated by a PCR-based method. The DNA sequences were verified by DNA sequencing. Baculoviruses were generated according to the manufacturer's

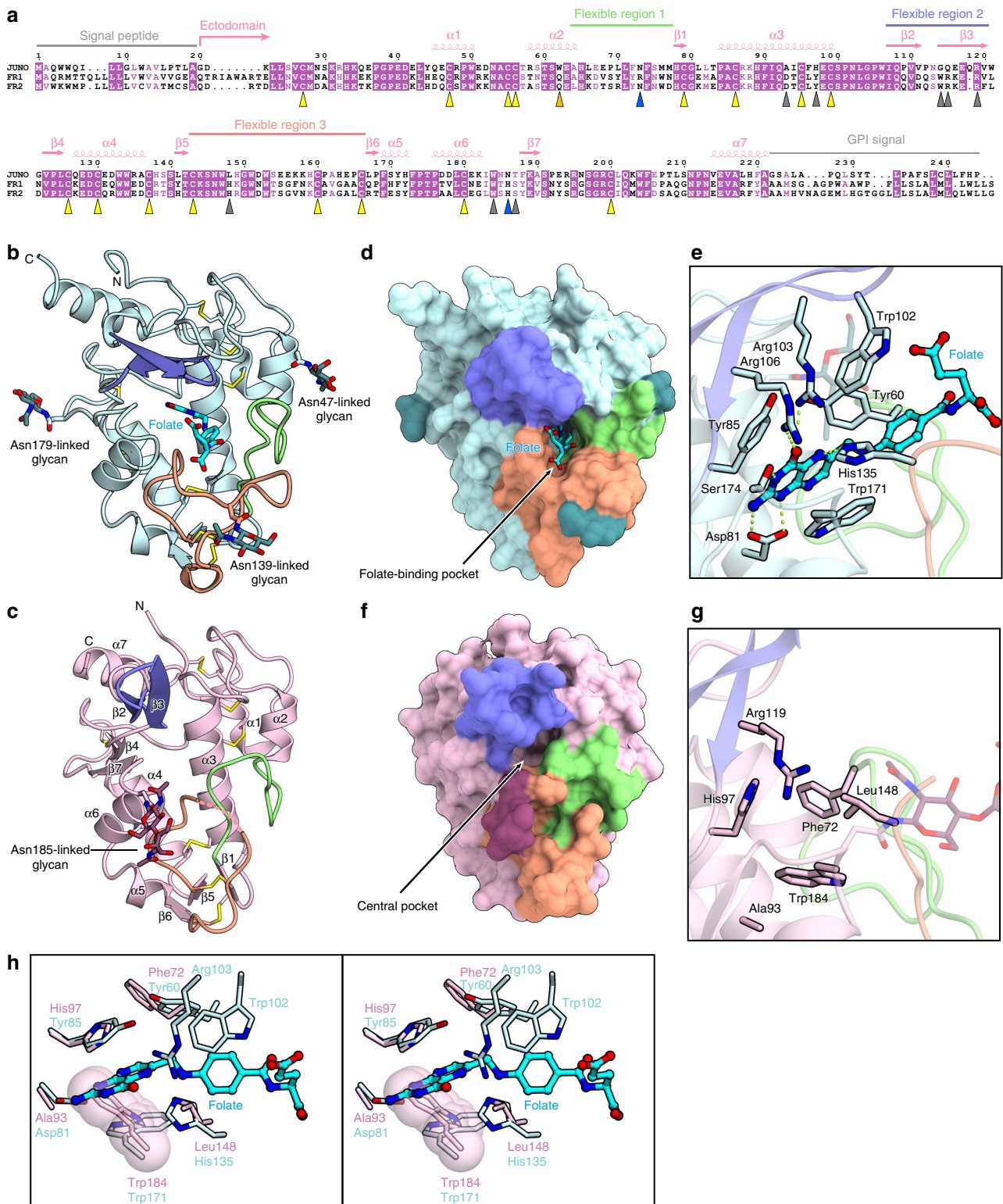

**Figure 2 | Comparison of JUNO with the FRs.** (**a**) Multiple sequence alignment of mouse JUNO, human FR1 and human FR2. The FR1 residues involved in folate binding are indicated by grey triangles. The other key residues are indicated as in Fig. 1a. (**b,c**) Ribbon representation of the human FR1–folate complex (PDB code 4LRH) (**b**) and N73D mutant JUNO (**c**). (**d,e**) Molecular surface (**d**) and folate-binding pocket (**e**) of the human FR1–folate complex (PDB code 4LRH). (**f,g**) Molecular surface (**f**) and central pocket (**g**) of N73D mutant JUNO. (**h**) Superimposition of the central pocket of N73D mutant JUNO and the folate-binding pocket of FR1 (stereo view). Trp184 of JUNO is highlighted as a semi-transparent space-filling model.

instructions, and baculovirus-infected Sf9 cells were cultured in Sf900II medium (Invitrogen) at 27 °C for 48 h before collecting. The culture supernatant was incubated with Ni Sepharose excel resin (GE Healthcare) at 4 °C overnight, and the resin was then washed with wash buffer (20 mM Tris-HCl, pH 7.5, 150 mM NaCl

and 20 mM imidazole). The bound protein was eluted with the wash buffer supplemented with 500 mM imidazole, and then dialysed against the wash buffer to remove the imidazole. The protein was incubated with Talon cobalt affinity resin (Clontech). The resin was washed with the wash buffer, and then the protein was

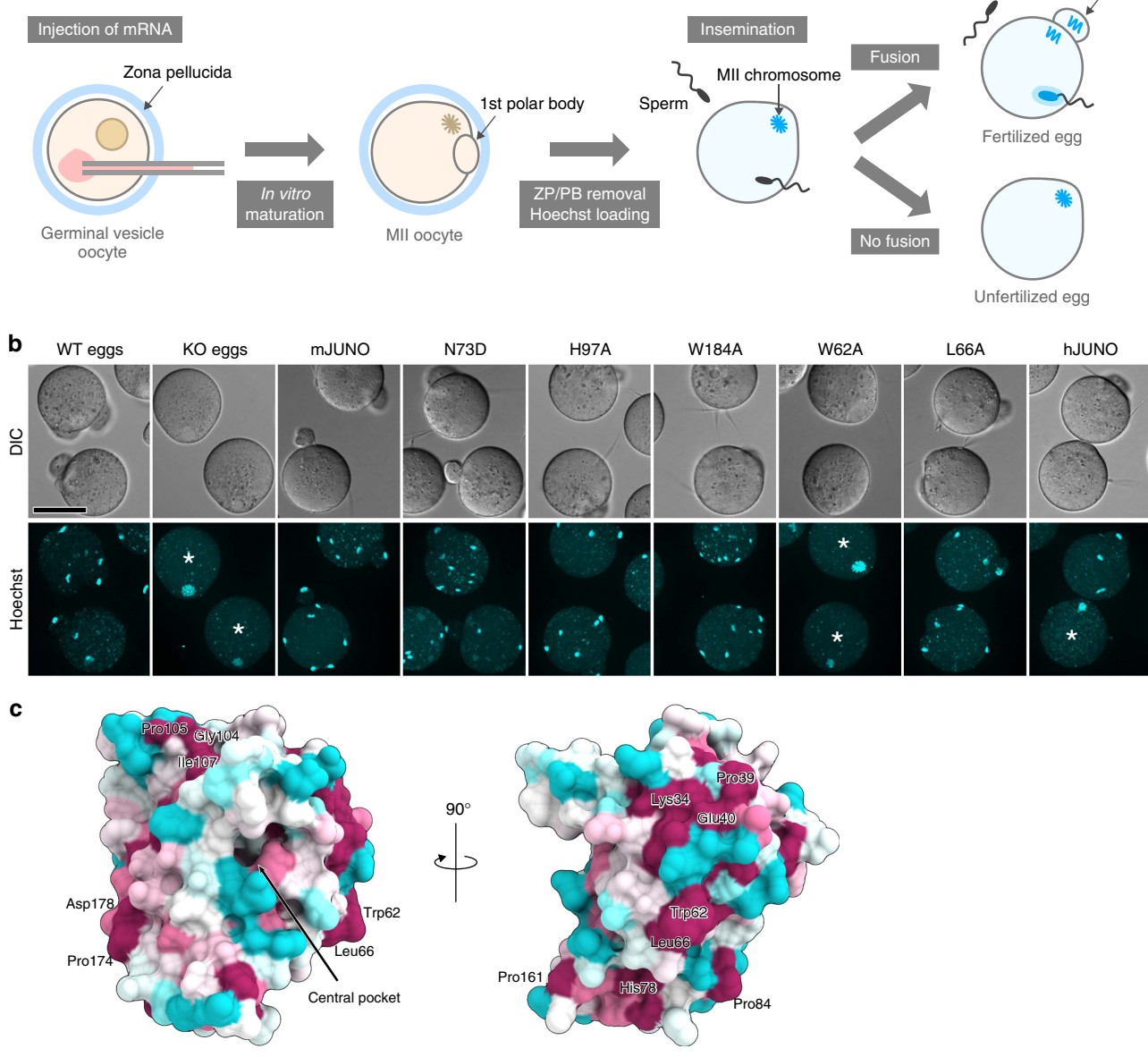

**Figure 3 | IZUMO1-binding site.** (**a**) Schematic showing the sperm–egg fusion complementation assay. After mRNA injection and *in vitro* maturation, Hoechst 33342 was loaded to visualize the MII chromosome and the fused sperm heads. (**b**) Functional complementation of *Juno* KO eggs by mRNA injection. The mRNA encoding the WT or mutants of mouse JUNO (mJUNO), or human JUNO (hJUNO), was injected into the mouse *Juno* KO eggs. The fusion of mouse spermatozoa was visualized by the transfer of DNA dye from the egg to the sperm nuclei. The extrusion of the second polar body was used as an indicator for successful fertilization. Asterisks indicate unfertilized eggs. Scale bar, 50 μm. (**c**) Conservation of the surface residues of the N73D mutant JUNO. The sequence conservation among JUNOs from 14 mammalian species (human, monkey, gorilla, chimpanzee, elephant, pig, horse, bear, cat, rabbit, golden hamster, mouse, rat and opossum) was calculated using the ConSurf server (http://consurf.tau.ac.il), and is coloured from cyan (low) to maroon (high).

eluted with the buffer supplemented with 300 mM imidazole. To cleave the C-terminal His tag, the eluted protein was mixed with the HRV3C protease and dialysed against buffer, consisting of 20 mM Tris-HCl, pH 7.5, 150 mM NaCl and 20 mM imidazole, at 4 °C overnight. The protein was passed through the Talon column again, and further purified by chromatography on Mono Q (GE Healthcare) and Superdex 200 columns (GE Healthcare). The purified N73D mutant JUNO protein was concentrated to 5 mg ml$^{-1}$, using an Amicon Ultra-4 filter (10 kDa molecular-weight cutoff; Millipore).

**Crystallography.** The purified N73D mutant JUNO protein was crystallized at 20 °C by the sitting drop vapour diffusion method. The crystals were obtained by mixing 0.1 μl of protein solution (5 mg ml$^{-1}$ JUNO, 10 mM Tris-HCl, pH 7.5, and 0.15 M NaCl) and 0.1 μl of reservoir solution (0.8 M ammonium phosphate dibasic, 0.1 M imidazole, pH 8.0 and 0.2 M NaCl). The crystals were cryoprotected in reservoir solution supplemented with 25% glycerol, and were

flash-cooled in a nitrogen stream. X-ray diffraction data were collected at 100 K on beamline BL32XU at SPring-8 (Japan). Diffraction data were processed with XDS[17] and Aimless[18]. The structure was determined by molecular replacement with Phaser[19]. The search model was built by the Phyre2 server[20], using the human FR1 (PDB code 4LRH) structure as the template. Model building and structural refinement were performed using COOT[21] and PHENIX[22], respectively. Data collection and refinement statistics are summarized in Table 1. Molecular graphic images were prepared using CueMol (http://www.cuemol.org).

**Animal experiments.** Neither randomization nor blinding method was used for the selection of animals. All animal experiments were approved by the Animal Care and Use Committee of the Research Institute for Microbial Diseases, Osaka University (Osaka, Japan).

**Generation of JUNO KO mice.** JUNO KO mice were generated as described previously[23,24]. In brief, EGR-G101 embryonic stem (ES) cells[25] (Riken BioResource Center: AES0182) were transfected with the pX330 vector, encoding humanized Cas9 and sgRNA targeted to the mouse *Juno* gene. The cell lines have been tested for mycoplasma contamination. The region containing the target sequence (GTGGGCAGTCCTACCCACCTTGG in exon 2) was amplified by PCR with the primers (5′-CCCCTGTTGGTCAGTTTGCTTTCTACATTC-3′ and 5′-GGAATGTTCCTCCTCCCACAGCC-3′), and then indels were determined by direct sequencing. The clone carrying indels in both alleles (10 bp deletion/1 bp insertion and 116 bp deletion) was injected into 8-cell ICR embryos, to generate chimeric mice. The germ line transmission of the 10 bp deletion (*Juno^em1Osb*) was confirmed by PCR and subsequent direct sequencing. The eggs from F2 female mice carrying 10 bp homozygous deletions (*Juno^em1Osb/em1Osb*: *Juno* KO) were immunostained with the anti-FR4 antibody (BioLegend: 125102) to monitor the loss of the JUNO protein, and were used in further studies.

**Generation of cDNAs and mRNAs encoding wild-type or mutant JUNOs.** The complementary DNA encoding mouse JUNO (GenBank accession number BC028431) was cloned between the *Xba*I and *Xho*I sites of the pCAGGS vector. The mutations were introduced by a PCR-based method. Human JUNO (GenBank accession number A6ND01) was also cloned into the pCAGGS vector. T7 promoter-tagged cDNAs were amplified by PCR with the primers (5′-GGGTA ATACGACTCACTATAGGGGCAACGTGCTGGTTGTTGTGC-3′ and 5′-AGCC AGAAGTCAGATGCTCAAGGGGCTTC-3′) and high fidelity Taq polymerase, using the pCAGGS vectors as templates. RNA synthesis, poly-A tail addition and further preparation for egg injection were performed as previously described[15,26],

**Table 2 | Fertilization rates in the sperm–egg fusion assay.**

| Egg genotype | mRNA | Number of experiments | Inseminated eggs | Fertilized eggs | (%) |
|---|---|---|---|---|---|
| WT | None | 4 | 31 | 29 | 93.5 |
| KO | None | 5 | 21 | 0 | 0.0 |
| KO | *mJuno* | 5 | 31 | 29 | 93.5 |
| KO | N73D | 3 | 19 | 18 | 94.7 |
| KO | H97A | 3 | 19 | 18 | 94.7 |
| KO | W184A | 3 | 18 | 17 | 94.4 |
| KO | W62A | 4 | 21 | 0 | 0.0 |
| KO | L66A | 3 | 18 | 15 | 83.3 |
| KO | *hJuno* | 3 | 24 | 12 | 50.0 |

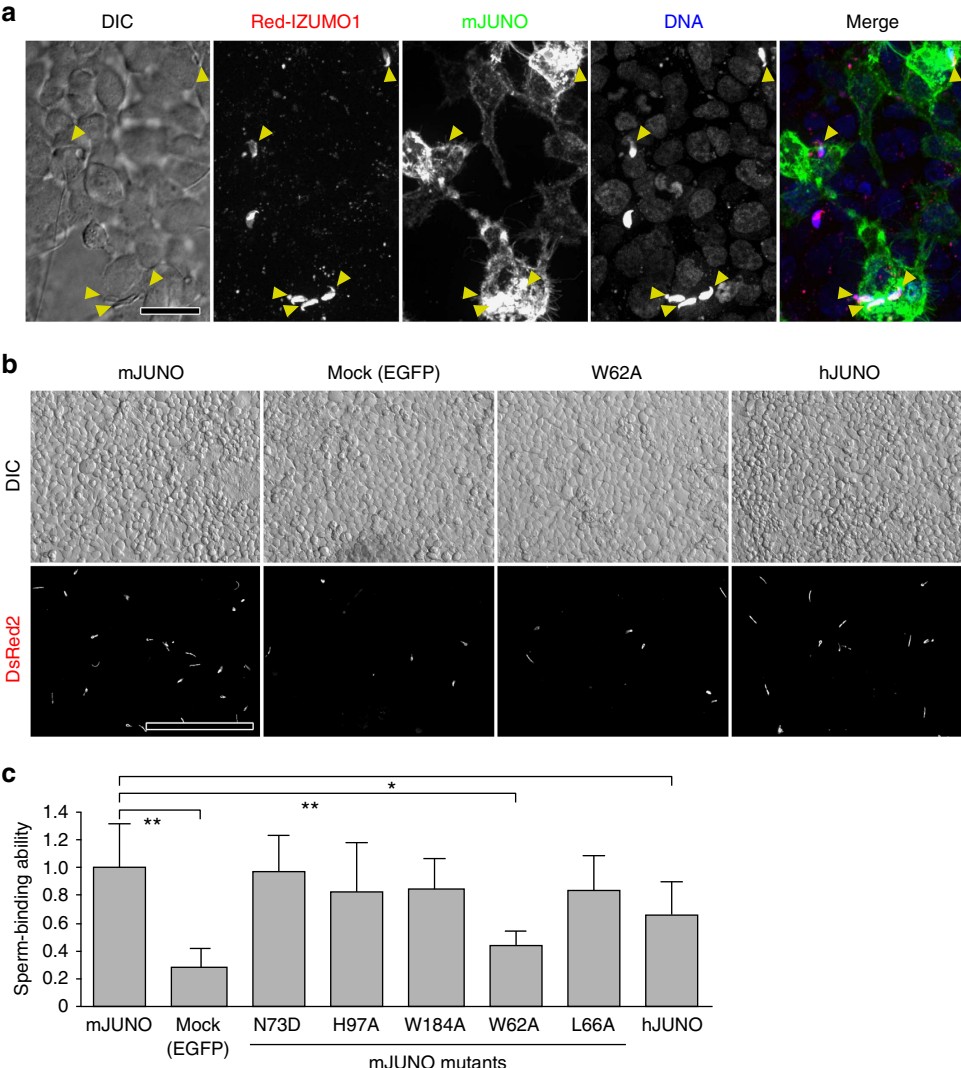

**Figure 4 | Sperm binding to JUNO-expressing cultured cells.** (**a**) Z-stack confocal microscopy image of seeded cells and inseminated Red-IZUMO1 spermatozoa. Red-IZUMO1, mJUNO and DNA (Hoechst 33342) signals are indicated in red, green and blue, respectively, in the merged image. The heads of spermatozoa bound to mJUNO-expressing HEK293T cells are indicated with arrowheads. Scale bar, 20 μm. (**b**) Binding of spermatozoa with mitochondrial DsRed2 to HEK293T cells expressing WT mJUNO, mutant mJUNOs or hJUNO. Light (upper image) and fluorescence (lower image) microscopic images of transfected cells after insemination with fluorescent spermatozoa are shown. Each picture corresponds to a quarter of the area of the field used for analysis. Scale bar, 200 μm. (**c**) Sperm-binding abilities of JUNO-expressing HEK293T cells. The number of spermatozoa bound to HEK293T cells expressing WT mJUNO was set to 1.0. Data are the mean ± s.d. Asterisks indicate significant differences (Student's *t*-test; *$P < 0.05$ and **$P < 0.005$).

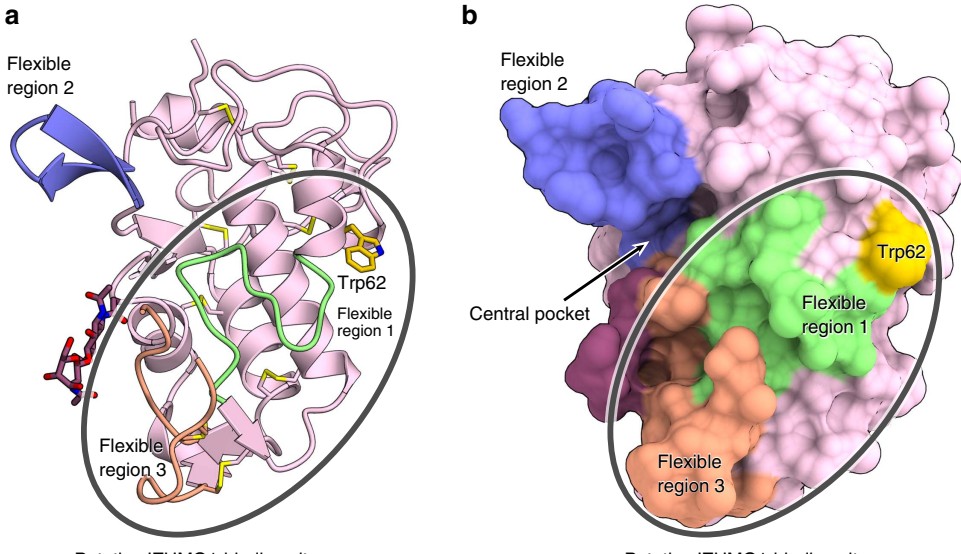

**Figure 5 | Putative IZUMO1-binding surface. (a,b)** A putative IZUMO1-binding site encompassing Trp62 and flexible regions 1 and 3 is indicated by a circle on the ribbon (**a**) and surface (**b**) representations of N73D mutant JUNO.

with slight modifications. Briefly, a mMESSAGE mMACHINE T7 kit (Ambion) and a Poly(A) Tailing kit (Ambion) were used, according to the manufacturer's instructions.

**Injection of wild-type or mutant *Juno* mRNAs into *Juno* KO eggs.** Immature GV (germinal vesicle) eggs were collected from the ovaries of wild-type B6D2F1 or *Juno* KO female mice (9–16 weeks old), 46 h after the injection of pregnant mare serum gonadotropin or anti-inhibin antibody (CARD HyperOva, Kyudo). Antral follicles, suspended in FHM medium supplemented with 250 μM dibutyryl-cyclic AMP, were punctured with 26G needles (Sigma). After dissociation of the cumulus cells by a glass pipette, the eggs were transferred to the medium supplemented with 5 μg ml$^{-1}$ cytochalasin B (Sigma), and mRNAs (50 ng ml$^{-1}$) were then injected into the eggs, using a piezo-micromanipulator with a glass capillary needle. The eggs were washed three times, and were cultured in TYH medium containing 10% fetal bovine serum (Biowest) for *in vitro* maturation. After 14 h, the zona were removed from the MII eggs, using a micromanipulator[27]. Pre-loading of Hoechst 33342 into the denuded eggs was performed, as previously described[2], and after three washes with fresh TYH medium, the eggs were subjected to mJUNO immunostaining or sperm–egg fusion assays. In one experiment, considering a failure of superovulation, 2 to 4 *Juno* KO female mice were assigned to an experimental group and 2 WT mice were assigned to a control group.

**Immunostaining of wild-type or mutant JUNOs on the cellular surface.** Eggs or HEK293T cells (Riken BioResource Center: RCB2202) were co-incubated for 30 min with the anti-FR4 antibody (BioLegend: 125102) and the Alexa Fluor 546-tagged secondary antibody (Invitrogen), under non-permeabilized conditions. After three washes, confocal images and epifluorescent images were acquired for the eggs and HEK293T cells, respectively. Confocal images were acquired using a spinning-disk confocal system quipped with a 30 × silicone oil immersion objective (Olympus)[15,28]. Z-stacks images were reconstructed from 120 images with a different z axis plane (1 μm increments), using the MetaMorph software (Molecular Devices). The HEK293T cell lines have been tested for mycoplasma contamination.

**Sperm–egg fusion assay.** Hoechst 33342 pre-loaded eggs were inseminated with 2 × 10$^5$ cells per ml B6D2F1 capacitated sperm for 30 min. After a quick wash in KSOM medium, the eggs were incubated for 30 min to allow the extrusion of the second polar body, and subjected to confocal imaging. Confocal images were acquired as described above. Experiments were repeated at least three times for each batch. Total numbers of inseminated eggs per batch (18–31) are indicated in Table 2.

**Sperm-binding assay.** HEK293T cells were seeded in six-well plates and transfected with the pCAGGS vectors encoding wild-type or mutant JUNOs, as previously described[24]. For mock transfection, pCAGGS vector encoding enhanced green fluorescent protein (cytosolic) was used. After 60 h, the cells were washed, incubated with 800 μl of TYH medium per well, and then mixed with capacitated spermatozoa from transgenic male mice carrying mitochondrial DsRed2 (ref. 29), at a final concentration of 2 × 10$^5$ cells per ml. After an incubation at 37 °C for

30 min, the cells were washed by gentle agitation in fresh medium for 15 min. Finally, the epifluorescent images of the DsRed2 signal were acquired for four randomly selected fields per well, using an inverted microscope equipped with a 10 × objective (IX70: Olympus), and were used for counting the number of spermatozoa. Experiments were repeated at least three times (total 12–16 fields per vector). The variances between groups are examined to be equal using *F*-test.

**Observation of the sperm–cell interaction.** Transfection of pCAGGS vector encoding WT JUNO, preparation of the HEK293T cells and insemination of Red-IZUMO1 transgenic spermatozoa were basically carried out as described[15] in sperm binding assay with slight modifications. Briefly, cells were seeded onto poly-L-lysine (Sigma) coated cover slip placed at the bottom of a six-well plate. Immunostaining of mJUNO and Hoechst 33342 staining were carried out during three washing steps (15 min in total). Confocal images were acquired with a 60 × silicone oil immersion objective (Olympus) and z-stacks images were reconstructed from 120 images with a different z axis plane (0.5 μm increments).

**Data availability.** Coordinates and structure factors have been deposited in the Protein Data Bank under accession code 5JYJ. All other data are available from the authors on reasonable request.

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

## Acknowledgements

We thank the beam-line staff at BL32XU and BL41XU at SPring-8 (Japan) for assistance with data collection. This work was supported by a grant from the Core Research for Evolutional Science and Technology (CREST) Program, The Creation of Basic Chronic Inflammation, from Japan Science and Technology Agency (JST) to O.N., and KAKENHI grant numbers 25112007 and 25250014 and a Takeda Science Foundation grant to M.I.

## Author contributions

K.K. prepared and crystallized the protein, collected the diffraction data, and solved the structure; Y.S. performed the egg–sperm fusion experiments; H.N. conceived the project and collected the diffraction data; A.K. constructed the mutant plasmids; J.M. assisted with the experiments; Y.F. and A.O. generated JUNO KO mice; H.N. and R.I. assisted with the structural analyses; K.K., Y.S., H.N., M.I. and O.N. wrote the manuscript with help from all authors. H.N., M.I. and O.N. supervised all of the research.

## Additional information

**Competing financial interests:** The authors declare no competing financial interests.

