## [Peer review file · Nature Communications]

Reviewers' comments:

Reviewer #1 (Remarks to the Author):

Manuscript No.: NCOMM516-04872-T

Title: Crystal structure of the egg Izumo1 receptor Juno

Authors: Kazuki, K., Satouh, Y., Nishimasu, H. et al. (Nureki, O.)

During mammalian fertilization two proteins have been shown to mediate binding of acrosome-reacted sperm to the plasma membrane of eggs - sperm Izumo1 and egg Juno. Izumo1 is a type-1 membrane protein associated with inner acrosomal sperm membrane and Juno is a GPI-anchored protein related to the folate receptor (FR) family of proteins (FR4) associated with egg plasma membrane. Binding of Izumo to Juno is essential for, but not solely responsible for, sperm-egg fusion (fertilization).

In this MS the authors describe the 3-dimensional structure of mouse Juno at 2.3 Å resolution and conclude that its overall structure is similar to other FRs, but it lacks the folate binding pocket observed in other FRs. This conclusion is identical to that of a very recent publication which described the 3-dimensional structure of mouse Juno at 2.7 Å resolution (Han, L. et al., *Current Biology* 26, R83-R101, 2016). In this publication the authors proposed that an alpha-helical region of Izumo1 is accommodated within the central pocket of Juno. However, the present MS proposes an alternative model for the interaction of Izumo1 and Juno in which Izumo1 interacts with a hydrophobic patch on the opposite side from the central pocket of Juno. Neither the recent publication nor present manuscript reports directly on the crystallographic structure of the Izumo1-Juno complex. Crystallographic analysis of the complex could resolve the differences between the two models proposed here and proposed by Han et al. in *Current Biology*, 2016.

The crystallographic data presented here appear to be very reliable. However, the authors used a Juno construct different from that used by previous authors (Han et al., *Current Biology*, 2016) and don't discuss the differences. They also do not discuss the similarities/differences between their Juno structure (2.7 Å) and the published structure of Juno (2.3 Å). Are the two 3-dimensional structures of Juno superimposable? Such comparisons would be very informative and essential for readers unused to dealing with 3-dimensional structures. In a similar vein, the authors leave any discussion of the published structure in *Current Biology* to the very end of their manuscript (reference 17); not a good idea, it should be up front in the MS and should be discussed.

The authors include some very interesting experiments in their MS using Juno KO mice (Fig. 3a,b,c) and somatic cells expressing Juno (Fig. 4), the most important of which is that mutation of conserved residues in the central pocket of Juno does not prevent complex formation between Juno and Izumo1. This finding suggests that the central pocket is not involved in binding to Izumo1 (i.e., contrary to the published conclusion by Han et al. in *Current Biology*, 2016). Rather, mutation of Trp62, a highly conserved residues among Junos, prevented formation of a Juno-Izumo1 complex and therefore binding of sperm to eggs (Figs. 3 and 4). The authors conclude that Trp62 of Juno interacts with hydrophobic residues on the alpha-helical core region of Izumo1. Again, this issue could be resolved by determination of the 3-dimensional structure of the Juno-Izumo1 complex. The authors carried out some additional experiments (Fig.7b) that confirm a previous finding (Bianchi et al., *Nature* 508, 483-487) that fusion of sperm and egg requires more than just formation of a Juno-Izumo1 complex. Presumably one or more "fusion proteins" are involved, but this is not discussed.

In conclusion, this is a very nice manuscript that describes a series of rigorous experiments that lead to some important conclusions about FRs, Juno, and the Juno-Izumo1 complex. The authors are to be complimented on their experimental approaches and interesting results. Again, this is a very nice piece of work. On the other hand, the authors could have done a much better job comparing their Juno structure to the published (*Current Biology*, 2016) Juno structure. As to which model of the Juno-Izumo1 complex is correct, the published model or the model presented

in this MS, we will have to wait for a direct crystallographic determination of the 3-dimensional structure of the complex.

Reviewer #2 (Remarks to the Author):

The paper by Kato et al. reports a 2.3 Å crystal structure of mouse Juno, the cysteine-rich GPA-anchored receptor on the egg surface that binds the sperm antigen IZUMO1 to initiate sperm-egg membrane fusion. Juno has high sequence homology to folate receptors and the overall fold of Juno is remarkably similar to that of folate receptors, with an rmsd of only 1 Å over 176 Cα atoms of FRα. Consistent with the inability of Juno to bind folate, several important folate-binding amino acids are altered in Juno and result in a position of W184 in the pocket that would clash with the position of folate in the structure alignment with FRα. To identify the IZUMO1-binding site on Juno, the authors generated CRISP/Cas9 Juno k.o. mice and Juno mutant mRNA in a fertilization complementation assay. While 2 conserved pocket mutants complemented the mutant fertilization defect, one of 2 tested residues (W62) in a hydrophobic patch failed to complement, yet was expressed. From this experiment the authors implied that the hydrophobic patch forms a key interaction interface. They further used a cell-based interaction assay to confirm that Juno-IZUMO1 binding is required for cell-cell interaction, but insufficient for membrane fusion, and that human Juno can interact with mouse IZUMO1. Overall, the paper reports an important structure of a key recognition protein for fertilization, which may have implications in the development of new contraceptives. While a similar structure has just been published in Current Biology (February 8, 2016), the novelty of the structure is still sufficiently high and the papers complement each other nicely to justify publication in Nature Communications, pending. However, some minor concerns need to be addressed prior to better understanding.

1. An R-free value of 0.28 for a resolution of 2.3 Å is quite high. Please provide Ramachandran statistics and a PDB validation report.
2. The statement in the Abstract that "...revealed that the conserved hydrophobic patch of Juno participates in IZUMO1 binding" is too strong. A single mutation is insufficient to identify a whole surface as interaction site. More appropriate would be something along the lines of "identified W62 as a residue that is required for full sperm binding and fertilization in cell-based assays". Even though W62 is surface exposed and the W62A mutant protein expressed on the cell surface, suggesting that it is indeed an interaction mutation, the authors cannot exclude an effect on the overall conformation.
3. The paper incorrectly cites Han et al. proposing that "IZUMO1 is accommodated within the central pocket of Juno, ...which is not consistent with our model". Instead, Han et al. identify the surface exposed flexible loops at the bottom of the pocket as a likely IZUMO1 binding site, which indeed has been confirmed by unpublished structures of the Juno-IZUMO1 complex.

Reviewer #3 (Remarks to the Author):

Sperm-egg fusion is a critical step in mammalian fertilization which is not elucidated yet. Izumo1 on the sperm membrane and its egg membrane receptor Juno are known to be essential. The intercellular bonds they form are necessary, even not sufficient for fusion. This study reports the crystal structure of mouse Juno and investigates the mode of Juno-Izumo1 interaction through the determination of the Juno's residue involved in its bond with Izumo1. Both aspects constitute valuable information for progressing in the understanding of sperm-egg interaction. However, two problems weaken the value of this work:

- 1- An article focusing on the same two aspects (crystal structure of mouse Juno and determination of the locus of interaction with Izumo1) has just been published (Han et al, Curr. Biol. CB 26, R100-101 (2016)). In itself, publishing a second paper on an identical topic is not a problem if the new data show divergences with the first ones and bring controversy. Apparently, this is the case here: (i) the crystal structure reported by the two studies does not seem to be identical, neither on the number of alpha helix, nor in the number of beta-sheets and disulfide bonds; (ii) in Han's paper, the complete ectodomain of mouse Juno was crystallized, while the authors of the present

manuscript had to make a mutation on one of the putative N-glycosylated site to be able to crystalize the protein;(iii) the location of the protein identified as involved in Juno/Izumo1 interaction is not the same in both cases. However, these differences are not or barely mentioned in the present paper. Han's paper is only cited in the conclusion of the manuscript while it should appear more like a starting point from which all the diverging results should be extensively discussed. Without this comparison and discussion, the authors miss the opportunity to defend the relevance for publishing a second paper reporting the crystal structure of Juno.

2- The determination of the residue involved in Juno/Izumo1 interaction and the demonstration of mouse and human Juno interchangeability regarding its interaction with mouse Izumo1 are strong points of this study. However, the authors need to support their conclusions with more readable images and convincing statistics: (i) The number of experiments relative to Figure 3, the number of inseminated eggs for each mutation, and the fertilization rate for each mutation must be given and discussed; (ii) The resolution of the pictures of Figure 7 is poor and the Figure caption does not describe properly the observations. In Figure 7d, the association between the white dots and the presence of sperm is not obvious. Each of the 8 fluorescent images should be accompanied with a zoom in fluorescence and DIC. Moreover these interesting results should be discussed and compared to Han's ones and an interpretation should be proposed concerning the divergences.

In conclusion, I would not recommend this manuscript for publication in Nature Communication in the present form. However I encourage the authors to re-structure it in order to highlight the controversy with previously published Juno crystal structure and defend the relevance of their data compared to previous ones.

Our point-by-point responses to the reviewers' comments are as follows:

Reviewer #1:

- The crystallographic data presented here appear to be very reliable. However, the authors used a Juno construct different from that used by previous authors (Han et al., Current Biology, 2016) and don't discuss the differences. They also do not discuss the similarities/differences between their Juno structure (2.7 Å) and the published structure of Juno (2.3 Å). Are the two 3-dimensional structures of Juno superimposable? Such comparisons would be very informative and essential for readers unused to dealing with 3-dimensional structures. In a similar vein, the authors leave any discussion of the published structure in Current Biology to the very end of their manuscript (reference 17); not a good idea, it should be up front in the MS and should be discussed

Thanks for your helpful comment. We have added the discussion on the comparison of our JUNO structure with the previously reported structures (Han et al. Curr. Biol. 2016) in the main text (Page 4). We have also added new figures showing the similarities and differences between our structure and their JUNO structures (Figure 1d and 1e). The structural comparison revealed that JUNO consists of a rigid core fold and three flexible loop regions. In particular, in the previous structures, the three flexible regions are largely disordered. Since the flexible loop regions were shown to be involved in IZUMO1 binding (Han et al. Curr. Biol. 2016), and are located in the proximity of Trp62, we proposed that this surface is a putative IZUMO1-binding site (Figure 5).

- The authors carried out some additional experiments (Fig.7b) that confirm a previous finding (Bianchi et al., Nature 508, 483-487) that fusion of sperm and egg requires more than just formation of a Juno-Izumo1 complex. Presumably one or more "fusion proteins" are involved, but this is not discussed.

Recently, Inoue et al. negated the direct participation of JUNO in cell-cell fusion processes, based on their behavior study showing that JUNO is excluded from the interactive surface at the moment of sperm-egg fusion, and postulated the involvement of one or more other factor(s) in the fusion process (Inoue et al. Nat. Commun. 2015). We included this information and discussed it in the revised manuscript (second paragraph in the Discussion).

Reviewer #2:

1. An R-free value of 0.28 for a resolution of 2.3 Å is quite high. Please provide Ramachandran statistics and a PDB validation report.

We have refined the structure to an R-free value of 0.23, and provided Ramachandran statistics in Table 1. The PDB validation report is attached to our revised manuscript.

2. The statement in the Abstract that "...revealed that the conserved hydrophobic patch of Juno participates in IZUMO1 binding" is too strong. A single mutation is insufficient to identify a whole surface as interaction site. More appropriate would be something along the lines of "identified W62 as a residue that is required for full sperm binding and fertilization in cell-based assays". Even though W62 is surface exposed and the W62A mutant protein expressed on the cell surface, suggesting that it is indeed an interaction mutation, the authors cannot exclude an effect on the overall conformation.

According to the reviewer's suggestion, we have revised the abstract to "Our complementation assays using *Juno* knockout eggs revealed that the conserved, surface-exposed tryptophan residue of JUNO is required for sperm binding and fertilization".

3. The paper incorrectly cites Han et al. proposing that "IZUMO1 is accommodated within the central pocket of Juno, ...which is not consistent with our model". Instead, Han et al. identify the surface exposed flexible loops at the bottom of the pocket as a likely IZUMO1 binding site, which indeed has been confirmed by unpublished structures of the Juno-IZUMO1 complex.

Thank you for your helpful comment. We agree with the reviewer, and found that the data from Han *et al.*, showing that the surface-exposed flexible loops are involved in IZUMO1 binding, are not inconsistent, but actually compatible with our model. Notably, the flexible loops are located near the Trp62 residue we identified as a key residue for IZUMO1 binding. Thus, in the revised manuscript, we concluded that the region encompassing the flexible loops and Trp62 is a putative IZUMO1-binding site (Figure 5).

Reviewer #3:

1-

(i) the crystal structure reported by the two studies does not seem to be identical, neither on the number of alpha helix, nor in the number of beta-sheets and disulfide bonds

(ii) in Han's paper, the complete ectodomain of mouse Juno was crystallized, while the authors of the present manuscript had to make a mutation on one of the putative N-glycosylated site to be able to crystalize the protein

We have added a discussion of the comparison of our JUNO structure with the previously reported structures (Han *et al. Curr. Biol.* 2016) (Page 4). We have also added new figures showing the comparison between our structure and their JUNO structures (Figure 1d and 1e). Although we

crystallized the N73D mutant of mouse JUNO, our structure is essentially identical to the wild-type structures (Han *et al. Curr. Biol.* 2016), suggesting that the N73D mutation has no substantial effect on the structure, consistent with our functional data. We have added the description in the revised manuscript.

(iii) the location of the protein identified as involved in Juno/Izumo1 interaction is not the same in both cases. However, these differences are not or barely mentioned in the present paper.

Han's paper is only cited in the conclusion of the manuscript while it should appear more like a starting point from which all the diverging results should be extensively discussed. Without this comparison and discussion, the authors miss the opportunity to defend the relevance for publishing a second paper reporting the crystal structure of Juno.

Thanks for your helpful comment. We have added the discussion about the comparison of our structure with the structures reported by Han *et al.* (Page 4). While Han *et al.* showed that the surface-exposed flexible loops are involved in IZUMO1 binding, we found that the conserved Trp62 residue participates in IZUMO1 binding. Since the flexible loops are located near the Trp62 residue, we concluded that the region encompassing the flexible loops and Trp62 is a putative IZUMO1-binding site, and proposed a revised model in the new figure (Figure 5).

2- The determination of the residue involved in Juno/Izumo1 interaction and the demonstration of mouse and human Juno interchangeability regarding its interaction with mouse Izumo1 are strong points of this study. However, the authors need to support their conclusions with more readable images and convincing statistics: (i) The number of experiments relative to Figure 3, the number of inseminated eggs for each mutation, and the fertilization rate for each mutation must be given and discussed; (ii) The resolution of the pictures of Figure 7 is poor and the Figure caption does not describe properly the observations. In Figure 7d, the association between the white dots and the presence of sperm is not obvious. Each of the 8 fluorescent images should be accompanied with a zoom in fluorescence and DIC. Moreover, these interesting results should be discussed and compared to Han's ones and an interpretation should be proposed concerning the divergences.

We increased the number of experiments and reconfirmed the reproducibility, and thus we updated the results (new Figure 3 and new Table 2). It should be noted that the fertilization rate with hJUNO is relatively lower than that with mJUNO, and the picture for the result was replaced with a more representative one (Figure 3b). We also replaced the pictures in the new Figure 4 (the previous Extended Data Figure 7) with new ones, and revised the figure legends. In addition, consistent with the results from Han *et al.*, we concluded that the region encompassing the flexible loops and Trp62 is a putative IZUMO1-binding site. These points are included and discussed in the revised manuscript.

Reviewers' comments:

Reviewer #1 provided no remarks to the Author and is satisfied with the revision.

Reviewer #2 (Remarks to the Author):

This reviewer feels a bit lost with the authors' rebuttal letter as commented directly following the authors' responses:

Reviewer #2:

1. An R-free value of 0.28 for a resolution of 2.3 Å is quite high. Please provide Ramachandran statistics and a PDB validation report.

We have refined the structure to an R-free value of 0.23, and provided Ramachandran statistics in Table 1. The PDB validation report is attached to our revised manuscript.

Table 1 of the merged PDF does NOT contain Ramachandran statistics and a PDB validation report is NOT attached.

2. The statement in the Abstract that "...revealed that the conserved hydrophobic patch of Juno participates in IZUMO1 binding" is too strong. A single mutation is insufficient to identify a whole surface as interaction site. More appropriate would be something along the lines of "identified W62 as a residue that is required for full sperm binding and fertilization in cell-based assays". Even though W62 is surface exposed and the W62A mutant protein expressed on the cell surface, suggesting that it is indeed an interaction mutation, the authors cannot exclude an effect on the overall conformation.

According to the reviewer's suggestion, we have revised the abstract to "Our complementation assays using Juno knockout eggs revealed that the conserved, surface-exposed tryptophan residue of JUNO is required for sperm binding and fertilization".

3. The paper incorrectly cites Han et al. proposing that "IZUMO1 is accommodated within the central pocket of Juno, ...which is not consistent with our model". Instead, Han et al. identify the surface exposed flexible loops at the bottom of the pocket as a likely IZUMO1 binding site, which indeed has been confirmed by unpublished structures of the Juno-IZUMO1 complex.

Thank you for your helpful comment. We agree with the reviewer, and found that the data from Han et al., showing that the surface-exposed flexible loops are involved in IZUMO1 binding, are not inconsistent, but actually compatible with our model. Notably, the flexible loops are located near the Trp62 residue we identified as a key residue for IZUMO1 binding. Thus, in the revised manuscript, we concluded that the region encompassing the flexible loops and Trp62 is a putative IZUMO1-binding site (Figure 5).

The authors still state "Based on these results, Han et al. proposed that the α -helical region of IZUMO1 is accommodated within the central pocket of JUNO. In contrast, our in vivo analyses revealed that Trp62, rather than the central pocket, is critical for sperm binding and fertilization,...

Juno has only one binding pocket, which is the one at the corresponding position of the ligand-binding pocket of folate receptors, and which is indeed labeled as "central pocket" in Fig. 2, 3, and 5 of the revised manuscript. Han et al. clearly state that they hypothesize that Izumo1 binds to the shallow surface groove surrounded by loops 1 and 3, and there is no mentioning of binding to the central pocket. Second, how can the authors claim that their identification of a single residue, W62, that is important for the interaction with Izumo1 excludes the importance of residues within flexible regions 1 and 3, unless they mutate surface residues of these regions. Instead, in addition to L66 adjacent to W62 in the "hydrophobic patch", they only mutate 2 residues that face into the central pocket.

It almost seems there must have been some errors in assembling the revised manuscript.

However, unless the points raised by this reviewer are appropriately addressed, I cannot recommend publication of the manuscript.

Reviewer #3 (Remarks to the Author):

This new version of the paper by Prof Nureki and co-workers entitled "Crystal structure of the egg IZUMO1 receptor JUNO" takes into account most of reviewer's comments. However, the authors do not manage yet to highlight the new message that distinguishes them from previous studies and especially Han's paper. The title itself is not judicious since it exclusively focuses on the crystal structure of Juno that has already been published by Han et al while the identification in Juno's structure of a binding site for Izumo1, which is probably the key point of this study, is not mentioned. Moreover in its present form, the title is misleading since it suggests that crystal structure of WT Juno was determined whereas N73D mutant Juno was in fact crystalized. The introduction lacks the presentation of the questions that the authors address in their manuscript. When discussed, the references to other papers sometimes lack accuracy and clear connection with the present data and results. The paragraph of the discussion that concludes the manuscript lacks rigor and some sentences are incomprehensible. I however still believe that this manuscript presents serious and convincing experiments that lead to important conclusions about Juno, its interaction with Izumo1, and the cross-species gamete recognition. Its publication under a more relevant and rigorous form would be beneficial to the scientific community.

Here are my detailed recommendations:

The title is not correct since "crystal structure of Juno" suggests that WT Juno was crystalized which is not the case in this study. Moreover, the title focuses on the crystal structure Juno that has already been published in its WT form. The title should reflect the new message of the article which is less the crystallization of mutant Juno than the identification of a binding site for Izumo1.

All over the manuscript, including in the legends of figures, mouse Juno is often used indifferently for WT-Juno or N73D mutant Juno which is a bit confusing. For more clarity, mouse Juno should systematically be replaced by WT-Juno or N73D mutant Juno

Abstract

L22 "triggered" is not rigorous in the present context. We just know that if interaction between Juno and Izumo1 cannot take place, subsequent sperm-egg fusion cannot occur. That does not mean that Juno-izumo1 interaction triggers fusion.

L25 same remark as L22 for "facilitates sperm-egg fusion". We cannot they that.

L30 "Our complementation assays" is not informative about the kind of experiments done.

Precise.

L32 Last sentence. Not relevant with the rest of the abstract.

Introduction

The crystal structure of Juno by Hans et al has to be already mentioned in the introduction as part of state of the art and the authors have to situate their work regarding previous ones.

The end of the introduction should present the questions the authors want to address and the way they tackle them.

L47= same remark as L25

Results

The text still does not clearly explain why the crystallization of WT Juno was possible for Han et al but not here.

L137 Title does not correspond to what is demonstrated in this paragraph. "Sperm binds to Juno

expressing cells and Trp62 of Juno is involved in this interaction", would be more correct.
L138-L139 The interpretation of ref 6 and 14 studies given in this sentence is not rigorous. What is demonstrated in ref 6 and 14 is that Juno and Izumo1 recognition is sufficient to generate cell-cell (or cell/sperm) adhesion but not cell-cell fusion .
L141 How are you sure that bound spermatozoa are acrosome reacted?
L143 In spite of "confirmed", "indicated" would be more correct, since we do not know what is confirmed.

Discussion

All the last paragraph of the discussion lacks rigor and some sentences are incomprehensible.
Line 182-182 The sentence is meaningless: cultured cells cannot be competent for sperm-egg-fusion.
L184-184. The sentence is not strictly correct. Ref 16 Figure 5e-f shows that during the sperm/egg interaction phase before the beginning of membrane fusion (Figure 5e) Juno is enriched in the sperm and egg interface, and a few minutes after fusion has started (Figure 5F) Juno is not anymore there while part of Izumo1 is still present suggesting that Izumo1 has another receptor that could be involved in membrane fusion.
Moreover the reference to Inoue's observation seems out of context of the present paper. Relevant observations and connections must be given and discussed
L186-189 the sentence is very complicated and the reader misses the message.
L189-192 The last two sentences do not bring any relevant information. Authors must be more explicit.

Figures

Fig 1 b What is the difference between the two ribbon representations?
Legend: precise that it is a representation of N73D mutant Juno.
Fig1 d - legend: precise that they are representation of WT-Juno and cite Han's paper.
Fig1 e- the legend should be replaced by "Superimposition of our N73D mutant Juno structure and the reported WT Juno structures (PDB code 5EJN) (gray)", and cite Han's paper for the latter.
Fig2 c and g legend Precise whether it is a representation obtained from N73D mutant Juno or WT-Juno
Fig 3a This figure is not very informative and its legend neither.
Fig3c Precise that the conservation is among JUNOs of "different species".
Fig 4 I do not find any definition for Mock(EGFP) neither in the figure legend, nor in the manuscript
Fig 4a inconsistency between the legend that indicates the use of Z-stack imaging and mat & meth section L429 & 430 that indicates that HEK cells were imaged in epifluorescence.

Reviewer #2:

1. An *R*-free value of 0.28 for a resolution of 2.3 Å is quite high. Please provide Ramachandran statistics and a PDB validation report.

We have refined the structure to an *R*-free value of 0.23, and provided Ramachandran statistics in Table 1. The PDB validation report is attached to our revised manuscript.

Table 1 of the merged PDF does NOT contain Ramachandran statistics and a PDB validation report is NOT attached.

We apologize for the mistakes. We have revised Table 1 and attached the PDB validation report.

2. The statement in the Abstract that "...revealed that the conserved hydrophobic patch of Juno participates in IZUMO1 binding" is too strong. A single mutation is insufficient to identify a whole surface as interaction site. More appropriate would be something along the lines of "identified W62 as a residue that is required for full sperm binding and fertilization in cell-based assays". Even though W62 is surface exposed and the W62A mutant protein expressed on the cell surface, suggesting that it is indeed an interaction mutation, the authors cannot exclude an effect on the overall conformation.

According to the reviewer's suggestion, we have revised the abstract to "Further complementation of Juno knockout eggs with mutant Juno mRNAs revealed that the conserved, surface-exposed tryptophan residue of JUNO is required for sperm binding and fertilization".

3. The paper incorrectly cites Han et al. proposing that "IZUMO1 is accommodated within the central pocket of Juno, ...which is not consistent with our model". Instead, Han et al. identify the surface exposed flexible loops at the bottom of the pocket as a likely IZUMO1 binding site, which indeed has been confirmed by unpublished structures of the Juno-IZUMO1 complex.

Thank you for your helpful comment. We agree with the reviewer, and found that the data from Han et al., showing that the surface-exposed flexible loops are involved in IZUMO1 binding, are not inconsistent, but actually compatible with our model. Notably, the flexible loops are located near the Trp62 residue we identified as a key residue for IZUMO1 binding. Thus, in the revised manuscript, we concluded that the region encompassing the flexible loops and Trp62 is a putative IZUMO1-binding site (Figure 5).

The authors still state "Based on these results, Han et al. proposed that the α -helical region of IZUMO1 is accommodated within the central pocket of JUNO. In contrast, our in vivo analyses revealed that Trp62, rather than the central pocket, is critical for sperm binding and fertilization,...Juno has only one binding pocket, which is the one at the corresponding position of the ligand-binding pocket of folate receptors, and which is indeed labeled as "central pocket" in Fig. 2, 3, and 5 of the revised manuscript. Han et al. clearly state that they hypothesize that Izumo1 binds to the shallow surface groove surrounded by loops 1 and 3, and there is no mentioning of binding to the

central pocket. Second, how can the authors claim that their identification of a single residue, W62, that is important for the interaction with Izumo1 excludes the importance of residues within flexible regions 1 and 3, unless they mutate surface residues of these regions. Instead, in addition to L66 adjacent to W62 in the "hydrophobic patch", they only mutate 2 residues that face into the central pocket.

We have revised the part of “Han *et al.* proposed that the α -helical region of IZUMO1 is accommodated within the central pocket of JUNO” to “Han *et al.* proposed that the α -helical region of IZUMO1 interacts with the surface groove surrounded by the two flexible loop regions of JUNO”.

Reviewer #3 (Remarks to the Author):

This new version of the paper by Prof Nureki and co-workers entitled "Crystal structure of the egg IZUMO1 receptor JUNO" takes into account most of reviewer's comments. However, the authors do not manage yet to highlight the new message that distinguishes them from previous studies and especially Han's paper. The title itself is not judicious since it exclusively focuses on the crystal structure of Juno that has already been published by Han et al while the identification in Juno's structure of a binding site for Izumo1, which is probably the key point of this study, is not mentioned. Moreover in its present form, the title is misleading since it suggests that crystal structure of WT Juno was determined whereas N73D mutant Juno was in fact crystalized. The introduction lacks the presentation of the questions that the authors address in their manuscript. When discussed, the references to other papers sometimes lack accuracy and clear connection with the present data and results.

The paragraph of the discussion that concludes the manuscript lacks rigor and some sentences are incomprehensible. I however still believe that this manuscript presents serious and convincing experiments that lead to important conclusions about Juno, its interaction with Izumo1, and the cross-species gamete recognition. Its publication under a more relevant and rigorous form would be beneficial to the scientific community.

Thank you for your helpful comments. According to your suggestion, we have revised our manuscript as follows:

Here are my detailed recommendations:

The title is not correct since "crystal structure of Juno" suggests that WT Juno was crystalized which is not the case in this study. Moreover, the title focuses on the crystal structure Juno that has already been published in its WT form. The title should reflect the new message of the article which is less the crystallization of mutant Juno than the identification of a binding site for Izumo1.

We have revised the title of our paper to “Structural and functional insights into IZUMO1 recognition by JUNO in mammalian fertilization”.

All over the manuscript, including in the legends of figures, mouse Juno is often used indifferently for WT-Juno or N73D mutant Juno which is a bit confusing. For more clarity, mouse Juno should systematically be replaced by WT-Juno or N73D mutant Juno

According to the reviewer’s suggestion, we have replaced mouse JUNO with WT JUNO or N73D mutant JUNO, paying attention not to change the meaning of original sentences.

Abstract

L22 "triggered" is not rigorous in the present context. We just know that if interaction between Juno and Izumo1 cannot take place, subsequent sperm-egg fusion cannot occur. That does not mean that Juno-izumo1 interaction triggers fusion.

Thank you very much for the comment. We replace “is triggered” with “requires”.

L25 same remark as L22 for "facilitates sperm-egg fusion". We cannot they that.

Following the reviewer’s advice, we rewrote the sentence as following; “Whereas other FRs bind and uptake folates, JUNO binds IZUMO1 and establishes the cell-cell adhesion.”

L30 "Our complementation assays" is not informative about the kind of experiments done. Precise.

We rewrote the sentence to be more precise and informative as following; “Further complementation of *Juno* knockout eggs with mutant *Juno* mRNAs revealed that the conserved, surface-exposed tryptophan residue of JUNO is required for sperm binding and fertilization.”

L32 Last sentence. Not relevant with the rest of the abstract.

Thank you very much for valuable comment, we re-wrote the last sentence to fit to the rest as following; “Our structure-based *in vivo* functional analyses provide a framework toward a mechanistic understanding of mammalian gamete recognition.”

Introduction

The crystal structure of Juno by Hans et al has to be already mentioned in the introduction as part of state of the art and the authors have to situate their work regarding previous ones. The end of the introduction should present the questions the authors want to address and the way they tackle them. L47= same remark as L25

According to reviewer's suggestion, we have added the description about the Han's previous study and revised our question in the end of Introduction.

Results

The text still does not clearly explain why the crystallization of WT Juno was possible for Han et al but not here.

Han *et al.* expressed WT JUNO using HEK293 GnT1⁻ cells as high-mannose glycosylated proteins followed by Endo H digestion, and successfully crystallized WT JUNO as a homogeneously glycosylated protein. In contrast, we expressed WT JUNO in insect cells as a heterogeneously glycosylated protein, and thus likely failed to crystallize WT JUNO. These results suggest that the glycosylation status of JUNO substantially affected the crystallization. We have added the description to the revised manuscript.

L137 Title does not correspond to what is demonstrated in this paragraph. "Sperm binds to Juno expressing cells and Trp62 of Juno is involved in this interaction", would be more correct.

According to the reviewer's suggestion, the title has been revised to "Involvement of JUNO Trp62 in the IZUMO1-JUNO-mediated cell-cell adhesion".

L138-L139 The interpretation of ref 6 and 14 studies given in this sentence is not rigorous. What is demonstrated in ref 6 and 14 is that Juno and Izumo1 recognition is sufficient to generate cell-cell (or cell/sperm) adhesion but not cell-cell fusion.

According to the reviewer's suggestion, the sentence has been revised.

L141 How are you sure that bound spermatozoa are acrosome reacted?

Thank you very much for comment. As explained in Satouh *et al.* JCS (2012), before acrosome reaction, the localization of Red-IZUMO1 is restricted to the acrosomal cap region, and, only after the acrosome reaction, Red-IZUMO1 or IZUMO1 distribute to other membrane compartments of the sperm head including the equatorial segment and postacrosomal region. To describe those, we added the explanation for it with the reference.

L143 In spite of "confirmed", "indicated" would be more correct, since we do not know what is confirmed.

According to the reviewer's suggestion, the sentence has been revised.

Discussion

All the last paragraph of the discussion lacks rigor and some sentences are incomprehensible.

Line 182-182 The sentence is meaningless: cultured cells cannot be competent for sperm-egg-fusion.

We apologize for the mistake. The sentence was revised as following; "Consistent with the previous studies, we showed that the cultured cells expressing mouse JUNO can bind to, but not fuse with spermatozoa (Fig. 4c)."

L184-184. The sentence is not strictly correct. Ref 16 Figure 5e-f shows that during the sperm/egg interaction phase before the beginning of membrane fusion (Figure 5e) Juno is enriched in the sperm and egg interface, and a few minutes after fusion has started (Figure 5F) Juno is not anymore there while part of Izumo1 is still present suggesting that Izumo1 has another receptor that could be involved in membrane fusion.

Moreover the reference to Inoue's observation seems out of context of the present paper. Relevant observations and connections must be given and discussed.

L186-189 the sentence is very complicated and the reader misses the message.

L189-192 The last two sentences do not bring any relevant information. Authors must be more explicit.

Thank you very much for your helpful comments. We carefully revised the whole paragraph, especially paying attention for both describing precise comprehension of previous studies and focusing on the relevant matters found in this study.

Figures

Fig 1 b What is the difference between the two ribbon representations?

Legend: precise that it is a representation of N73D mutant Juno.

Fig. 1 b is a stereo image as indicated in the parenthesis in the figure title. We have revised the figure legend, according to the reviewer's suggestion.

Fig1 d - legend: precise that they are representation of WT-Juno and cite Han's paper.

Fig1 e- the legend should be replaced by "Superimposition of our N73D mutant Juno structure and the reported WT Juno structures (PDB code 5EJN) (gray)", and cite Han's paper for the latter.

According to the reviewer's suggestion, the figure legend and labels in the figure panel have been revised.

Fig2 c and g legend Precise whether it is a representation obtained from N73D mutant Juno or WT-Juno

According to the reviewer's suggestion, the sentences have been revised.

Fig 3a This figure is not very informative and its legend neither.

Thanks for the comment. We revised both the figure and the legend to explain our assay more precisely.

Fig3c Precise that the conservation is among JUNOs of "different species".

We calculated surface conservation between JUNO proteins from 14 mammalian species. We have added the description about how to calculate the conservation in the figure legend.

Fig 4 I do not find any definition for Mock(EGFP) neither in the figure legend, nor in the manuscript

We newly added the description for that part in Materials and Methods.

Fig 4a inconsistency between the legend that indicates the use of Z-stack imaging and mat & meth section L429 & 430 that indicates that HEK cells were imaged in epifluorescence.

The methods for the observation of Red-IZUMO1 spermatozoa on seeded cells was described in different part (L444-446 in the previous manuscript). To avoid confusion, we revised whole the Materials and Methods section to describe what type of observation were carried out in each assays with any microscopies.

REVIEWERS' COMMENTS:

Reviewer #2 (Remarks to the Author):

The remaining points raised by this reviewer have been satisfactorily addressed in the new version.

Reviewer #3 (Remarks to the Author):

I'm satisfied with authors' revision, except for:

- line 60 in the introduction: authors still write "crystal structure of Juno" instead of the crystal structure of N73D mutant Juno.

- Methods section

Protein preparation: the preparation of the ectodomain of WT Juno is given, but not that of N73D mutant protein

Crystallography: line 383 replace Juno by N73D mutant Juno

- Table 1: replace Mouse Juno by N73D mutant Juno

Our point-by-point responses to the reviewers' comments are as follows:

Reviewer #2:

The remaining points raised by this reviewer have been satisfactorily addressed in the new version.

Thank you very much for the comment.

Reviewer #3 (Remarks to the Author):

I'm satisfied with authors' revision, except for:

- line 60 in the introduction: authors still write "crystal structure of Juno" instead of the crystal structure of N73D mutant Juno.

- Methods section

Protein preparation: the preparation of the ectodomain of WT Juno is given, but not that of N73D mutant protein

Crystallography: line 383 replace Juno by N73D mutant Juno

- Table 1: replace Mouse Juno by N73D mutant Juno

We revised the manuscript according to the suggestions.